# Study of Genotoxicity, Activities on Caspase 8 and on the Stabilization of the Topoisomerase Complex of Isoeleutherin and Analogues

**DOI:** 10.3390/molecules28041630

**Published:** 2023-02-08

**Authors:** Kelly Cristina Oliveira de Albuquerque, Natasha Costa da Rocha Galucio, Gleison Gonçalves Ferreira, Ana Carolina Sousa Quaresma, Valdicley Vieira Vale, Marcelo de Oliveira Bahia, Rommel Mario Rodriguez Burbano, Fábio Alberto de Molfetta, Sandro Percario, Maria Fâni Dolabela

**Affiliations:** 1Postgraduate Program in Biodiversity and Biotechnology (BIONORTE), Federal University of Pará, Belém 66075-110, Brazil; 2Postgraduate Program in Genetics and Molecular Biology, Federal University of Pará, Belém 66075-110, Brazil; 3Laboratory of Pharmacology of Neglected Diseases, Health Sciences Institute, Federal University of Pará, Belém 66075-110, Brazil; 4Postgraduate Program in Pharmaceutical Sciences, Institute of Health Sciences, Federal University of Pará, Belém 66075-110, Brazil; 5Postgraduate Program in Pharmaceutical Innovation, Federal University of Pará, Belém 66075-110, Brazil; 6Postgraduate Program in Neuroscience and Cell Biology, Federal University of Pará, Belém 66075-110, Brazil; 7Laboratory of Molecular Modeling, Institute of Exact and Natural Sciences, Federal University of Pará, Belém 66075-110, Brazil

**Keywords:** *Eleutherine plicata*, genotoxicity, isoeleutherin, topoisomerase II, toxicity

## Abstract

This study evaluated the genotoxicity of Ethanol Extract (EEEp), Dichloromethane Fraction (FDCMEp) and isoeleutherin isolated from *Eleutherine plicata*, using the micronucleus test and the impact of structural alterations on toxicity and molecular docking (topoisomerase II and DNA complex). The extract was obtained by maceration and fractionation in a chromatography column. The genotoxicity was evaluated by the micronucleus test in human hepatoma cells (HepG2). Isoeleutherin was the starting molecule in the search for analogues by structural similarity, using the ZINC and e-Molecules databases. Isoeleutherin and analogues were subjected to in silico toxicity prediction, and compounds free of toxicological risks (CP13, CP14, CP17 and isoeleutherin) were selected for molecular docking in Topoisomerase II (PDB: 1ZXM). In the micronucleus test, isoeleutherin was less genotoxic. Among the 22 isoeleutherin analogues there were variations in the toxicity profile. Molecular docking studies showed that the compounds have good complementarity in the active site with important hydrogens bonds. Therefore, the structural changes of isoeleutherin led to the obtaining of a molecule with a lower mutagenic potential, and the CP13 can be considered a prototype compound for the development of new molecules with pharmacological potential.

## 1. Introduction

*Eleutherine plicata* Herb. (Iridaceae) is a native American plant, with reports of occurrence in other tropical countries and regions such as Southern Africa, the eastern Mediterranean, and Central and South America. In Brazil, it is popularly known as marupari, marupazinho, marupá-piranga, palmeirinha, lírio-folha-de-palmeira and wáro. It presents itself in the form of a clump with red bulbs similar to onions, features whole, simple and pleated leaves with colorful flowers from white to pink. This species is widely used in Brazilian popular herbal medicine [1,2].

*Eleutherine plicata* Herb. has as heterotypic synonym *Eleutherine bulbosa* (Mill.) Urb, *Cipura plicata* (Sw.) Griseb, *Eleutherine americana* (Aubl.) Merr. ex K. Heyne, *Eleutherine anomala* Herb, *Eleutherine longifolia* Gagnep, *Eleutherine subaphylla* Gagnep, *Ferraria parviflora* Salisb, *Galatea americana* (Aubl.) Kuntze, *Galatea plicata* (Sw.) Baker, *Galatea vespertina* Salisb, *Ixia americana* Aubl, Marica plicata (Sw.) Ker Gawl, *Moraea plicata* Sw, *Sisyrinchium americanum* (Aubl.) Lemée, *Sisyrinchium capitatum* Pers, *Sisyrinchium elatum* Seub. *ex* Klatt, *Sisyrinchium latifolium* Sw, *Sisyrinchium plicatum* (Sw.) Spreng, *Sisyrinchium racemosum* Pers, *Bermudiana bulbosa* (Mill.) Molina, *Galatea bulbosa* (Mill.) Britton and basionym *Sisyrinchium bulbosum* Mill [3].

The species is widely used in popular medicine in almost all regions of Brazil, with the heaviest usage in the Amazon region, due in large part to the ethnopharmacological dimensions of indigenous societies to which its first uses are related. *E. plicata* has popular claims as an anthelmintic, a menstrual regulator and abortifacient, and for disorders of the gastrointestinal tract. Scientifically, antimicrobial, healing, purgative, abortive, cardiovascular and respiratory system activities have been proven [4].

The classes of metabolites isolated from *E. plicata* include anthraquinones (mainly chrysophanol and steroidal sapogenin types), organic acids, alkaloids, aurones, coumarins, chalconam flavonoids and, primarily, naphthalene, anthraquinones and naphthoquinones. These metabolites are listed in the List of Medicinal Plants of Interest to SUS–Brazilian Unified Health System (RENISUS) [5].

This species has been shown to be an important source of chemicals such as naphthoquinones, a class of natural products with various biological activities. The following compounds have already been isolated from this species: isoeleutherin (Figure 1A), eleutherin (Figure 1B), eleutherol (Figure 1C), eleutherinone (Figure 1D), (R)-4-Hydroxyeleutherine (Figure 1E), eleuthrone (Figure 1F), isoeleuthoside C (Figure 1G) and eleutherinol-8-O-b-D-glucoside (Figure 1H) [2,6,7,8]. Some studies demonstrated the activity of extracts, fractions and/or isolated substances from *E. plicata* against malaria parasites [7] and as antimicrobial [9] and antitumor agents [10]. Pharmacological activities have been related to quinone compounds, especially isoeleutherin [7,10].

Samples of *E. plicata* were submitted to the micronucleus assay and we observed that the fractionation of the ethanol extract from *E. plicata* bulbs (EEEp) led to a fraction with higher toxic potential (Dichloromethane Fraction (FDCMEp)). Eleutherin and isoeleutherin were isolated from this fraction and proved to be less toxic than FDCMEp. The mutagenic potential test of eleutherin and isoleutherin in the *Allium cepa* model demonstrated that eleutherin has a higher mutagenic potential [8]. Another study evaluated the acute oral toxicity of EEEp, FDCMEp and isoeleutherin that did not cause significant clinical and laboratory changes in mice [11]. In order to understand the mechanism involved in cell death caused by isoeleutherin and eleutherin, molecular docking and dynamics were performed in caspase 8, which is a protein involved in apoptosis. The study demonstrated these compounds stably bind to this protein [11].

In another study, the isoeleutherin and eleutherin ability to stabilize the DNA-topoisomerase II complex was evaluated and the stabilization of the complex was confirmed [8]. This stabilization may be involved in DNA fragmentation [12,13,14].

The present study evaluated the genotoxicity of EEEp, FDCMEp and isoeleutherin using the micronucleus test. In addition, we performed an *in silico* study of isoeleutherin analogues, evaluating the impact of structural changes in toxicity and molecular docking in the stabilization of topoisomerase II and DNA (TOPO II) complex.

## 2. Results and Discussion

### 2.1. Chemistry

From the ethanolic extract of *E. plicata* (yield = 8.65%) the dichloromethane fraction was obtained (yield = 15.66%), in which the presence of naphthoquinone was detected by thin layer chromatography (TLC). The fractionation of FDCMEp led to the isolation of three subfractions (S1, S2 and S3), with S2 being identified as isoeleutherin. Other studies of this species, carried out by the same group, also isolated and identified isoeleutherin [7,8,10].

Naphthoquinones are a major group found in *E. plicata* and are associated with the biological activities described for the species. They are quinones associated with the naphthalene system, a general group characterized by the presence of two carbonyl groups formed together by a conjugated system of two double bonds (C-C). Naphthoquinones are associated with the cytotoxic potential linked to the species, especially isoeleuterine through the inhibition of transcription factors [5,7,8,10].

### 2.2. Micronucleus Assay

EEEp, FDCMEp and Isoeleutherin were evaluated by micronucleus frequency using human hepatoma cells (HepG2). The fractionation of EEEp contributed to the increase in frequency. FDCMEp presented higher rates than EEEp and isoeleutherin with lower toxicity (Table 1).

However, a different study evaluated the genotoxicity of these samples using the comet assay and the same cell line, with the highest rate of damage observed in cells treated with isoeleutherin [11].

Studies that evaluated the contribution of fractionation of the EEEp extract in relation to cytotoxicity in VERO cells obtained results of CC_50_= 28.71 μg/mL + 1.054. The FDCMEp presented CC_50_= 26.19 μg/mL + 1.670. When comparing the results obtained, an increase in cytotoxicity was observed, suggesting a higher concentration of naphthoquinones in the FDCMEp. The FDCMEp subfractions resulted in CC_50_= 14.36 μg/mL + 1.12 (S1) and CC_50_= 18.67 μg/mL + 1.25 (S2), indicating that the fractionation contributed positively to the cytotoxic activity [11].

Another study, using the *Allium cepa* method, showed the rate of chromosomal aberrations caused by isoeleutherin was lower than the positive control, colchicine [8]. In summary, there are controversial results regarding isoeleutherin. There are studies that indicate its low genotoxic potential, for example, the present study [8] while there are others that demonstrate genotoxicity [11]. In order to better understand the toxicity of isoeleutherin and whether structural changes could interfere with it, prediction studies were carried out for its analogues.

### 2.3. In Silico Studies to Predict the Toxicity of Isoeleutherin and Analogues

Studies aimed at predicting the toxicity of drug candidates are an important phase of development and cannot be excluded. Many drugs are withdrawn from the market due to toxic metabolites generated by these substances [15]. Over the years, some computational tools have been created to facilitate and streamline the drug development process, with a more rational design focused on known targets, thereby making the process faster and cheaper [15,16,17].

Isoleutherin and analogues were submitted to toxicity prediction studies in different models, including: Algae, Daphnia, Medaka, Minnow, cytotoxicity, mutagenicity, carcinogenicity and HeRG. The structural alterations of isoeleutherin do not seem to change the toxicity (Table 2), suggesting the toxicity of these molecules is related to the quinone group. This group is often mediated by the presence of cyclohexadienedione structure, being mediators of toxicity and/or cytoprotection, related to the oxidative metabolism of these compounds [18]. 

Biological activities of naphthoquinones, present in extracts and fractions of *E. plicata*, have been related to a redox potential of these constituents. Free radicals have one or more unpaired electrons, making them unstable and highly reactive. Examples of free radicals are: the hydrogen atom, superoxide (O2•-), hydroxyl (OH•) and nitric oxide (NO•). Oxidation is a fundamental part of the aerobic pathway and metabolism; thus, free radicals are produced naturally. These free radicals, whose unpaired electron is centered on oxygen or nitrogen atoms, are called reactive oxygen species (ROS), or reactive nitrogen species (RNS) [22,23], respectively. In the body, free radicals are involved in phagocytosis, regulation of cell growth, intercellular signaling and the synthesis of biological substances. However, an excess of free radicals causes harmful effects, such as lipid peroxidation of membranes and protein oxidation [24].

Species such as *E. plicata*, *E. americana* and *Cipura paludosa* are described in the literature as an important source of naphthoquinones. These species present high levels of this class of natural products and diverse biological activities. Aqueous extract obtained from dried bulbs of these herbs has been described as generating oxidative stress by inducing the deleterious endogenous formation of a bioactive oxygen-derived species that promotes apoptosis, inhibits inflammation and has important anticancer properties in different cancer cell lineages, such as glioma (U-251), breast (MCF-7), ovary (NCI/ADR-RES), kidney (786-0), non-small cell lung (NCI-H460), colon (HT-29), HepG2 cells and leukemia (K562) [11,25].

Regarding acute oral toxicity, some compounds were classified as class 3 (CP7, 10, 11 and 16), others as class 4 (CP1, 2, 3, 4, 5, 6, 7, 8, 9, 12, 13, 14, 15, 17 and isoeleutherin) and some do not have data available to allow such analysis (CP18, 19, 20 and 21; Table 3). The CP7, 10, 11 and 16 have the CH2-COOH radical in common. This is a structural difference that seems to favor acute oral toxicity (Table 3).

A positive point is that the molecules do not have hepatoxic potential (Table 3). It is known that many drugs undergo hepatic metabolism. As such, changes in liver function can interfere with their metabolism, this being one of the main causes of safety-related failures in drug development. Early identification represents a strategy to eliminate this risk [26]. Another very interesting point is that some molecules do not seem to have immunotoxic potential (CP18, 20 and 21; Table 2). When a compound is immunotoxic, it can interfere with several signaling pathways, resulting in changes in cytokine production and marker expression [27].

### 2.4. Molecular Docking

To assess whether the docking parameters of the GOLD program are satisfactory for describing the macromolecular target and the ligand conformation, a re-docking was performed using the co-crystallized ligand (ANP) in the TOPO II enzyme (PDB code: 1ZXM). This target was selected because isoeleutherin stabilizes the DNA-TOPO II complex and this could be involved in cell death mechanism [8]. The value of the Root Mean Square Deviation (RMSD) between the experimental and the docked (ANP) using the Fconv program was 0.452 Å. When the RMSD values are less than 2 Å, the model achieves satisfactory solutions and reproduces the orientation of the co-crystallized ligand [28]. Thus, the GOLD program showed a good ability for predicting the orientation of the crystallographic ligand in the active site of the TOPO II enzyme.

Based on the toxicity prediction studies, we selected CP13, CP14, CP17 and isoeleutherin to perform the TOPO II molecular docking study. From the values of the scores obtained in the GOLD program, it was observed that the compound CP13 presented the highest value (59.27), followed by CP14, CP17 and isoeleutherin, which presented values of 52.76, 48.45 and 44.15, respectively.

From Figure 2, it can be seen that CP13 was the selected compound that presented the greatest number of hydrogen bonds, and this compound presented hydrogen bonds with Asp94, Thr147, Asn150 and Lys168 residues. The CP14 made hydrogen bonds with Asp94 and Asn150 residues, while CP17 did not make hydrogen bond interactions in the active site. Isoeleutherin, on the other hand, made a hydrogen bond with the residue Ile141. It is observed that these results corroborate those obtained by Alam et al. in a study with the topoisomerase II enzyme, in which the compounds formed hydrogen bonds with the catalytic residues Ile141, Thr147, Asn150 and Lys168, essential for enzymatic activity [29].

## 3. Material and Methods

### 3.1. Plant Material, Extract and Fractions

The *E. plicata* bulbs were collected in Vila Fátima, a municipality of Traquateua in the state of Pará, Brazil (BR 318, Lat. 1.1436°, Long. 46.95511°) The botanical identification was deposited in the Herbarium João Murça Pires, of the Museu Paraense Emílio Goeldi under the registration MG. 202631. Subsequently, the bulbs were washed in running water, cut into smaller pieces and subjected to the drying process in a circulating air oven. After drying, they were ground in knife mills to obtain the powder and subjected to maceration in 96° GL ethanol. Then, the extractive solution was concentrated, resulting in the EEEp [8,10].

The EEEp was subjected to fractionation in column chromatography with silica gel as the stationary phase, resulting in 4 fractions: hexane, dichloromethane, ethyl acetate and methanol. The dichloromethane fraction (FDCMEp) was subjected to fractionation by preparative thin layer chromatography, obtaining 3 subfractions (S1, S2 and S3). The S2 fraction was identified by NMR as isoeleutherin.

### 3.2. Selection of Isoeleutherin Analogues

The reference molecule in this work, isoeleutherin, was used to search for analogues through structural similarity in the ZINC [30] and e-Molecules databases (Figure 3) [31].

### 3.3. In Silico Studies to Predict Toxicity

A combination of online tools (PreADMET and ProTox-II) was used to predict various toxicity outcomes such as acute toxicity, hepatotoxicity, cytotoxicity, carcinogenicity, mutagenicity and immunotoxicity. These tools use the structure of the ligand to correlate with extant drugs and toxic compounds and with known toxicity. In this study, the predictions evaluated the similarities for mutagenic activity, toxicological dosage and the level for different tissues in different models. Pharmacologically relevant properties of the compounds were predicted. The models employed here, PreADMET and ProTox-II, are tools based on the Quantitative Structure–Activity Relationship (QSAR) where the molecular structure is related to the effects on ecosystems and principally on human or microbiological metabolism. To this end, the QSAR model performs two processes. First, it correlates the chemical structure to a biological activity from existing products and then predicts values related to possible action through mathematical regression calculations. Calculations are based on variable predictions (X) and variable potential responses (Y). Among the primarily analyzed molecular properties, understood here as descriptors, are the physical–chemical and topographical characteristics of the molecules that can be applied in 2D or 3D, being used by us for the 2D model. The speed afforded by these tools allowed us to characterize what was observed in the in vitro study, whether isoeleutherin and its analogues hypothetically presented toxicity in the studied models and whether their use could lead to unwanted organic dysfunctions based on the relationship between their chemical structure and known toxic groups. The method is widely used to select molecules with an accurate potential to transform into drugs [32].

### 3.4. Molecular Docking

Molecular docking was used to explore the possible conformations of the ligand with the binding receptor, estimating the strength of protein–ligand interaction [33]. Isoeleutherin and its analogues were obtained from SMILES files and downloaded from the ZINC [30] and eMolecules [31] databases. The crystallographic structure of the TOPO II enzyme was retrieved from the Protein Data Bank (PDB) under the code 1ZXM [14] with a resolution of 1.87 Å. Molecular docking simulations were performed using the Genetic Optimization for Ligand Docking (GOLD) program, version 2020.1, which uses a genetic algorithm to generate and select conformations of flexible compounds that bind to a protein receptor [34]. The GOLD program considers the flexibility of the molecule and amino acid residues at the receptor site, and is an efficient tool for discovering the binding mode of new ligands [35]. Furthermore, Janežič et al. carried out a virtual screen study with a collection of natural compounds with the TOPO II enzyme using the GOLD program, in which they showed that flavonoid derivatives act as catalytic inhibitors of this enzyme [36]. First, to validate the docking protocol, a re-docking was carried out with the co-crystallized ligand (ANP—phosphoaminophosphonic acid-adenylate ester) [14], and the protocol that reached conformations with RMSD values below 1 Å in relation to the structure experimental was selected to perform the docking simulations. The RMSD values were calculated using the Fconv program [37]. Prior to the docking simulation, the enzyme was prepared by removing water molecules and ligands and by adding hydrogen atoms. The active site was defined with a radius of 10 Å of the ANP ligand and the isoeleutherin derivatives were docked in the GOLD program, applying the GoldScore and ChemScore scoring functions [38] with 100% search efficiency. Analysis of intermolecular interactions’ calculations were performed using Discovery Studio Visualizer (Dassault Systèmes BIOVIA, Discovery Studio Modeling Ambiente, version 2021, San Diego: Dassault Systemes, 2022) [39].

### 3.5. Micronucleus Technique with Cytokinesis Block (CBMN, for Cytokinesis-Block Micronucleus)

The micronucleus test was performed with HepG2 cells, distributed in a 12-well plate at a concentration of 2 × 10^5^ cells/mL in RPMI-1610 medium supplemented with 10% fetal bovine serum at a temperature of 37 °C and a humidified atmosphere containing 5% CO_2_ for 20h. After this period, the cells were treated with the following concentrations of samples: EEEp (9.8 µg/mL, 4.9 µg/mL and 2.45 µg/mL), FDCMEp (9.5 µg/mL, 4.75 µg/ mL and 2.375 µg/mL) and isoeleutherin (7.685 µg/mL, 3.842 µg/mL and 1.921 µg/mL). The negative control consisted of cells and culture medium and the positive control was composed by cells treated with doxorubicin (0.02 µg/mL), due to its previously known genotoxicity. After a period of 24h, 3 µg/mL of cytochalasin-B was added and 24 h later the cells were trypsinized and centrifuged at 1000 rpm/5min, the supernatant was discarded and 5 mL of ice-cold hypotonic solution (0.075 M KCl) was added. Later, they were fixed with methanol and acetic acid for the preparation of slides, dried at room temperature and stained with 5% Giemsa [40]. The analysis of several parameters, such as the conventional micronucleus and the Cell Division Index (CDI), was performed in an optical light microscope at 1000x magnification.

For micronucleus analysis, 1000 binucleated cells were selected with intact nuclei and membrane. Those chosen were of identical morphology to the main nuclei, with a diameter between 1/16 to 1/3 of the main nuclei. The micronuclei could not have refringence or be connected to one main nuclei; they should have the same coloring as the nuclei; and could be adjacent to, but not superimposed on the nuclei. The Nuclear Division Index (NDI) was calculated by the following formula:IDM = [M1+ 2(M2) + 3(M3) + 4(M4)]/N(1)
where the proportion of cells with 1, 2, 3 and 4 nuclei (M1 to M4) in 500 viable cells was verified (N) [41].

## 4. Conclusions

Isoeleutherin was the compound that showed the lowest cytotoxicity in the micronucleus assay; therefore, it did not present toxicological risks. The structural changes suggest an increase in affinity with the TOPO II enzyme, observed in the increase in the amount of hydrogen bond interactions performed with amino acid residues of the active site. Using an integrative approach of in vitro and in silico studies to better understand the genotoxicity of isoeleutherin, we obtained molecular results from docking that suggest a good association with the biological activity observed for TOPO II, demonstrating that these compounds can be a good starting point in the search for new drugs for anticancer therapy.

## Figures and Tables

**Figure 1 molecules-28-01630-f001:**
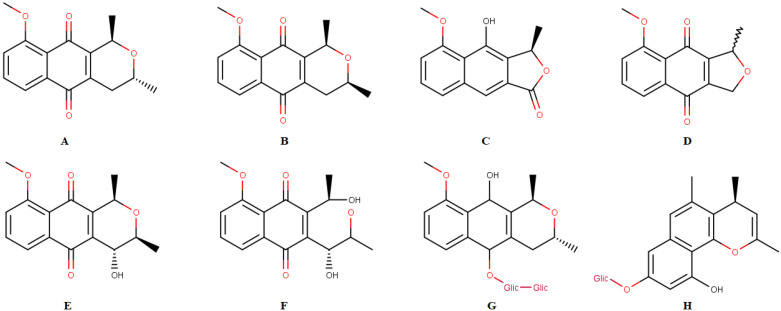
Chemical constituents isolated from *Eleutherine plicata*: isoeleutherin (**A**), eleutherin (**B**), eleutherol (**C**), eleutherinone (**D**), (R) 4-Hydroxyeleutherine (**E**), eleuthrone (**F**), isoeleuthoside C (**G**), eleutherinol-8-O-β-D-glucoside (**H**).

**Figure 2 molecules-28-01630-f002:**
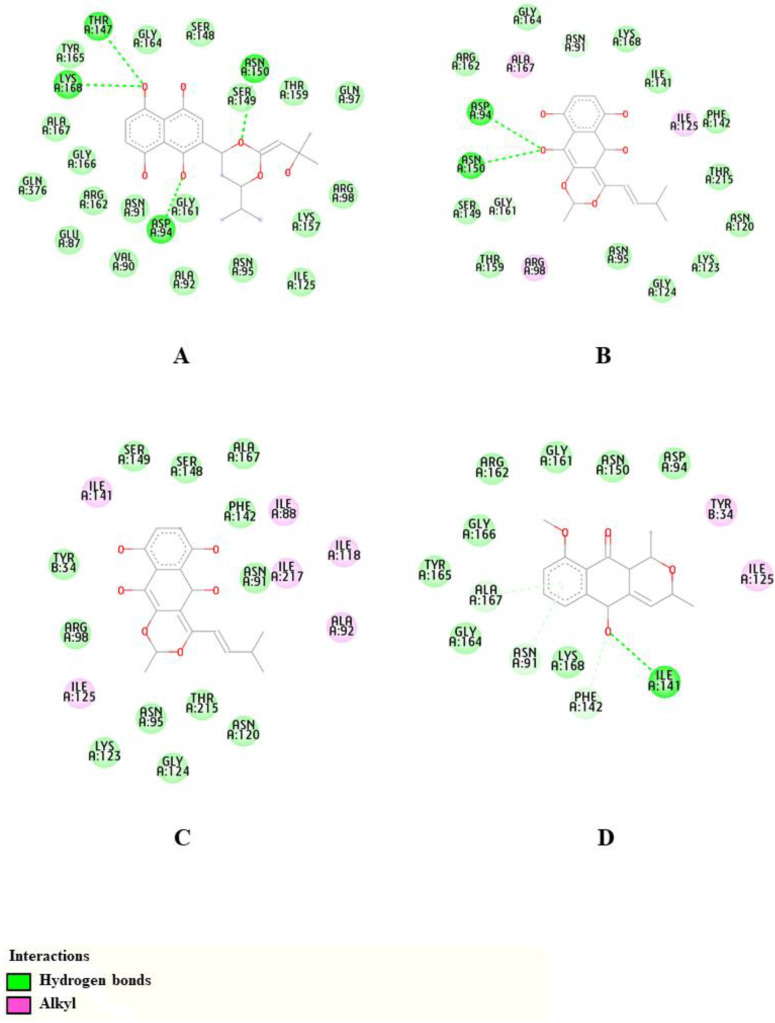
The 2D representations of intramolecular interactions of compounds CP13 (**A**), CP14 (**B**), CP17 (**C**) and Isoeleutherin (**D**) with Topoisomerase II.

**Figure 3 molecules-28-01630-f003:**
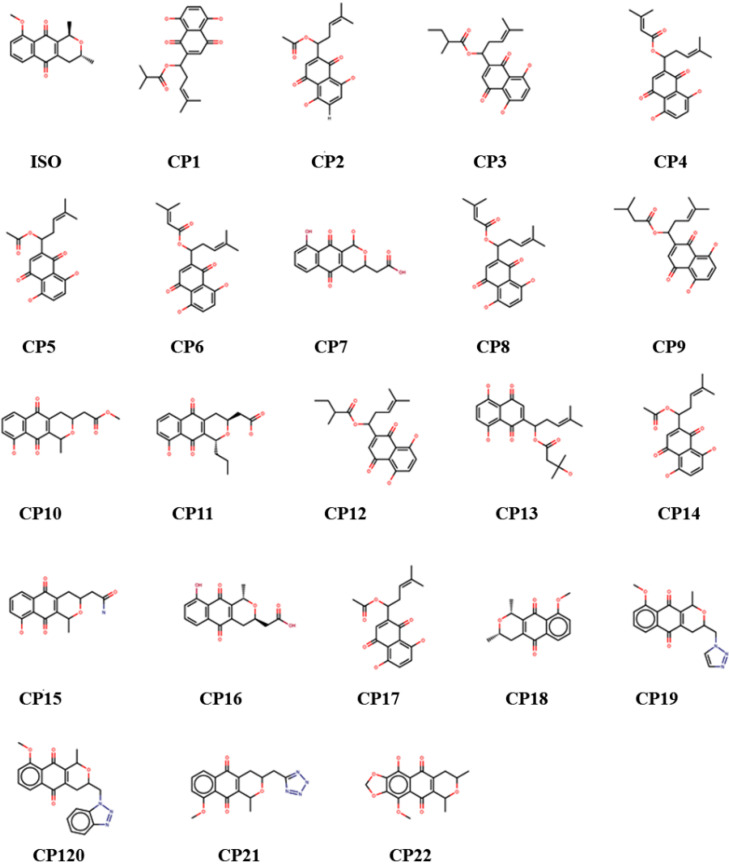
Chemical structure of isoeleutherin and analogues.

**Table 1 molecules-28-01630-t001:** Micronucleus frequency (MF) and nuclear division index (NDI) caused by *E. plicata* samples in HepG2 cells.

Samples	Concentrations (µg/mL)	MF (%) ± SD	NDI (%) ± SD
Negative control	-	2.40 ± 0.56	1.94 ± 0.01
Positive control (Doxorubicin)	0.02	36.07 ± 1.36	1.87 ± 0.05
EEEp	9.80 4.90 2.45	9.13 ± 0.35 5.30 ± 0.53 2.87 ± 0.36	1.34 ± 0.03 1.43 ± 0.02 1.65 + 0.08
FDCMEp	9.52 4.76 2.38	20.43 ± 0.70 6.83 ± 0.31 5.20 ± 0.30	1,20 ± 0.02 1.24 ± 0.01 1,36 ± 0.02
Isoeleutherin	15.55 7.77 3.88	4.03 ± 0.25 2.90 ± 0.20 2.10 ± 0.36	1.36 ± 0.16 1.39 ± 0.15 1.51 ± 0.15

SD = standard deviation.

**Table 2 molecules-28-01630-t002:** Isoeleutherin analogues and their toxicity.

	Toxicity *				Protox		Mutagenicity *	Carcinogenicity *		Protox	HeRG *
	Algae	Daphnia	Minnow	Medaka	Cytotoxicity	Mutagenicity		Mouse	Rat	Carc.	
CP1	T	T	VT	VT	Toxic	NM	M (0)	+	+	-	LR
CP2	T	T	VT	VT	Toxic	NM	M (0)	+	+	-	LR
CP3	T	T	VT	VT	Toxic	NM	M (0)	+	+	-	LR
CP4	T	T	VT	VT	Toxic	NM	M (1)	+	+	-	MR
CP5	T	T	VT	VT	Toxic	NM	M (0)	+	+	-	LR
CP6	T	T	VT	VT	Toxic	NM	M (1)	+	+	-	MR
CP7	T	NT	VT	VT	Non-toxic	NM	M (1)	-	+	+	LR
CP8	T	T	VT	VT	Toxic	NM	M (0)	+	+	-	MR
CP9	T	T	VT	VT	Toxic	NM	M (0)	+	+	-	LR
CP10	T	NT	VT	VT	Non-toxic	NM	M (1)	-	+	+	LR
CP11	T	T	VT	VT	Non-toxic	NM	M (1)	-	+	+	LR
CP12	T	T	VT	VT	Toxic	NM	M (0)	+	+	-	LR
CP13	T	T	VT	VT	Non-toxic	NM	NM (0)	+	-	-	LR
CP14	T	T	VT	VT	Toxic	NM	M (0)	+	+	-	LR
CP15	T	NT	VT	VT	Non-toxic	NM	M (2)	-	+	+	LR
CP16	T	NT	VT	VT	Non-toxic	NM	M (1)	-	+	+	LR
CP17	T	T	VT	VT	Toxic	NM	M (0)	+	+	-	LR
CP18	T	NT	VT	VT	Non-toxic	NM	M (2)	-	+	+	MR
CP19	T	NT	VT	VT	Non-toxic	M	M (2)	-	+	-	MR
CP20	T	T	VT	VT	Non-toxic	M	M (2)	-	-	+	MR
CP21	T	NT	VT	VT	Non-toxic	M	M (2)	-	+	-	LR
CP22	T	NT	VT	VT	Non-toxic	NM	M (2)	-	+	+	LR
ISO	T	T	VT	VT	Non-toxic	NM	M (1)	-	+	+	MR

* Preadmet: Toxicity in algae (T-toxic < 1mg/mL and NT-non-toxic > 1 mg/mL) [19]; Daphnia sp (T-toxic < 0.22 µg/mL and NT-non-toxic > 0.22 µg/mL) [20], Medaka (VT-very toxic < 1 mg/L, T-toxic between 1–10 mg/L, harmful between 10–100 mg/L and NT-non-toxic > 100 mg/L) [21]; Minnow (VT—Very toxic < 1mg/L; T—Toxic 1–10 mg/L; ST—Slightly toxic 10–100 mg/L; Non-Toxic > 100mg/L) [19]; Carcinogenicity: C: Carcinogenic; NC: Non-carcinogenic and Mutagenicity: M: Mutagenic; NM: Non-Mutagenic; LR: low risk; MR: medium risk; M (0/1/2): number of Salmonella strains that were positive.

**Table 3 molecules-28-01630-t003:** Toxicity of isoleutherin and analogues.

	Acute Oral Toxicity		Hepatotoxicity	Immunotoxicity
	DL_50_ (mg/kg)	Class		
CP1	1000	4	-	+
CP2	1000	4	-	+
CP3	1000	4	-	+
CP4	1000	4	-	+
CP5	1000	4	-	+
CP6	1000	4	-	+
CP7	290	3	-	+
CP8	1000	4	-	+
CP9	1000	4	-	+
CP10	290	3	-	+
CP11	290	3	-	+
CP12	1000	4	-	+
CP13	1000	4	-	+
CP14	1000	4	-	+
CP15	1000	4	-	+
CP16	290	3	-	+
CP17	1000	4	-	+
CP18	NE	NE	-	-
CP19	NE	NE	-	+
CP20	NE	NE	-	-
CP21	NE	NE	-	-
CP22	NE	NE	-	+
ISO	1000	4	-	+

Class 3: toxic if swallowed (50 < LD_50_
≤ 300 mg/kg); 4: Hazardous if swallowed (300 < LD_50_
≤ 2000 mg/kg); NE-not evaluated.

## Data Availability

Data are available from the corresponding author upon request.

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
