# Peer review of "Study of Genotoxicity, Activities on Caspase 8 and on the Stabilization of the Topoisomerase Complex of Isoeleutherin and Analogues"

_molecules, 2023, doi:10.3390/molecules28041630_

Round 1

Reviewer 1 Report

In this manuscript, authors evaluated the genotoxicity of EEEp, FDCMEp and isoeleutherin isolated from 23 Eleutherine plicata, using the micronucleus test and the impact of structural alterations on toxicity 24 and molecular docking (topisomerase II and DNA complex).  The genotoxicity was evaluated by the micro-26 nucleus test in human hepatoma cells (HepG2). Authors found that Isoeleutherin was the compound that showed the lowest cytotoxicity in the 234 micronucleus assay. This is an interesting work, after a minor revision, it can be published in Molecules.

Authors should give the  lowest binding energies in the molecular mocking.

Author Response

Dear reviewer, We greatly appreciate your encouragement to our work. About your placement:
Authors should give the  lowest binding energies in the molecular docking.
On page 3, lines 145 and 146, the values of the score function are described, GoldScore, Gold program standard.  

Reviewer 2 Report

The present article explores the study of genotoxicity, activities on caspase 8 and on the stabilization of the topoisomerase complex of isoeleutherin and analogues. Overall, the manuscript is well-prepared and organized. This topic is interesting and I recommend it to publish in this journal after minor revisions.

1.      In the abstract, first you should write the full form of these (EEEp, FDCMEp) later you can use the standard abbreviation (Line 23, page 1).

2.      Eleutherine plicata should be in italics (Line 24, page 1).

3.      Figure 01 should be written as Figure 1, and it should be bold in line 51, page 2.

4.      You can cite this paper also.

Quadros Gomes AR, da Rocha Galucio NC, de Albuquerque KCO, Brígido HPC, Varela ELP, Castro ALG, Vale VV, Bahia MO, Rodriguez Burbano RM, de Molfeta FA, Carneiro LA, Percario S, Dolabela MF. Toxicity evaluation of Eleutherine plicata Herb. extracts and possible cell death mechanism. Toxicol Rep. 2021 Jul 31;8:1480-1487. doi: 10.1016/j.toxrep.2021.07.015. PMID: 34401358; PMCID: PMC8353407.

5.      Illustrate what are, in your opinion, the manuscript’s strengths and weaknesses.

Author Response

Dear reviewer, We appreciate the corrections proposed to improve our article. About your placements: 1. In the abstract, first you should write the full form of these (EEEp, FDCMEp) later you can use the standard abbreviation (Line 23, page 1).
Realized 2. Eleutherine plicata should be in italics (Line 24, page 1).
Realized
3. Figure 01 should be written as Figure 1, and it should be bold in line 51, page 2. Realized
4. You can cite this paper also.
Quadros Gomes AR, da Rocha Galucio NC, de Albuquerque KCO, Brígido HPC, Varela ELP, Castro ALG, Vale VV, Bahia MO, Rodriguez Burbano RM, de Molfeta FA, Carneiro LA, Percario S, Dolabela MF. Toxicity evaluation of Eleutherine plicata Herb. extracts and possible cell death mechanism. Toxicol Rep. 2021 Jul 31;8:1480-1487. doi: 10.1016/j.toxrep.2021.07.015. PMID: 34401358; PMCID: PMC8353407. This suggested article is being used as a reference in the article, inserted in the list of references as number 8. Page 2, line 62 and page 5, line 276.

5. Illustrate what are, in your opinion, the manuscript’s strengths and weaknesses.
The strong point of the article was to demonstrate the impact of fractionation, of extracts obtained from plant species from the Amazon, in the assessment of toxicity. Likewise, verify whether structural changes in isoeleutherine can cause changes in toxicity.
The weak point of the article is that no compound was completely devoid of toxicity.